# Digital Addiction Intervention for Children and Adolescents: A Scoping Review

**DOI:** 10.3390/ijerph20064777

**Published:** 2023-03-08

**Authors:** Keya Ding, Hui Li

**Affiliations:** 1Shanghai Institute of Early Childhood Education, Shanghai Normal University, Shanghai 200235, China; keya@shnu.edu.cn; 2Macquarie School of Education, Macquarie University, Sydney, NSW 2109, Australia

**Keywords:** digital addiction, psychological interventions, digital-based intervention, children and adolescents

## Abstract

Digital devices play a significant role in the learning and living of children and adolescents, whose overuse or addiction has become a global concern. This scoping review seeks to synthesize existing studies to investigate relevant interventions and their effects on digital addiction in children (ages 0–18). To understand the latest advances, we have identified 17 studies published in international peer-reviewed journals between 2018–2022. The findings revealed that, first, most interventions for digital addiction in children and adolescents were cognitive–behavioral therapies (CBT) or CBT-based interventions, which could improve anxiety, depression, and related symptoms of digital addiction. Second, rather than directly targeting addictive behaviors, some family-based interventions aim to strengthen family functions and relationships. Finally, digital-based interventions, such as website-based, application-based, and virtual reality interventions, are promising in adolescent digital addiction interventions. However, these studies shared the same limitations: small sample sizes, short intervention durations, no control group, and nonrandomized assignments. The small sample size problem is difficult to solve by offline intervention. Meanwhile, online digital-based intervention is still in its infancy, resulting in limited generalizability of the findings and the inability to popularize digital intervention. Accordingly, future intervention studies should integrate various assessments and interventions to form an integrated platform to provide interventions for addicted children and adolescents worldwide.

## 1. Introduction

Various easy-to-carry digital devices (smartphones, tablets, laptops, etc.) have emerged rapidly since the turn of this millennium, driving human beings into the “digital era”. These digital devices have become daily necessities for human learning and life, influencing children’s studies, entertainment, and social interactions. Since 2007 when Apple launched the first generation of the iPhone, smartphones and tablets have evolved dramatically and thoroughly changed the landscape of digital use. Digital devices are no longer bound by time or place; digital media are boundless everywhere. Children could use them anywhere, anytime, wandering on social media, surfing the Internet, and playing video games. As young minds are premature and thus very vulnerable, young children tend to become addicted to playing mobile phones, video games, and social media, causing the phenomenon of digital addiction (“DA” hereafter). In general, DA refers to any “addictive behaviors” associated with using digital devices such as mobile phones, computers, the Internet, video games, and social media [1]. In order to provide a common ground for diagnostic criteria and a call for further research, the American Psychiatric Association finally included “Internet gaming disorder (IGD),” which includes both online and offline gaming, in Section III of the DSM-5 in 2013 [2]. Following the World Health Organization’s recognition of IGD as a disease, the “gaming disorder” diagnosis was added to the ICD-11 in 2018 [3]. According to these criteria, more and more children have become digital addicts, especially during the COVID-19 pandemic. DA has thus become a pressing public health issue of global concern, and some intervention programs have been developed and launched to cope with this public crisis. However, no systematic reviews have been conducted to synthesize the intervention studies during the recent five years to identify effective programs and to inform policymaking and practices. This scoping review aims to fill this gap by synthesizing the relevant interventions and their effects during 2018–2022.

### 1.1. Digital Addiction: A Global Health Problem

In many countries, internet addiction has become a serious problem for mental health. A meta-analysis of 31 countries revealed that the global Internet use disorder prevalence was estimated at 6.0% among 12–41-year-olds, with the Middle East having the highest plurality [4]. Another 3-decade meta-analysis found a global IGD prevalence of 4.6% among adolescents aged 10 to 19 years [5]. Furthermore, data from multiple studies indicate that Internet use disorder is more common in younger age groups. For example, the Canadian National Center for Addiction and Mental Health polled 11,438 students in grades 7–12. They discovered that 20% of students spend 5 h or more daily on social media, 23% play video games almost every day, and 30% use various electronic devices. Approximately 5% of these students reported signs of addiction [6]. As early as 2013, 70% of adolescents (ages 14–18) used social media daily, and by 2018, 45% of adolescents spent time online “almost all the time” [7]. Furthermore, digital devices are used at younger and younger ages. A recent Chinese study reported that 96.5% and 89.4% of families with children aged 1–3 owned smartphones and tablet PCs, respectively, and nearly half (47.4%) of them used smartphones at least once a day, while about a third (36.8%) used tablets at least once a day [8].

### 1.2. Digital Addiction Hurts Children and Adolescent Health

It was widely reported that DA could cause significant distress and functional impairments in daily life, as well as comorbid psychiatric disorders, such as attention deficit and hyperactivity disorder, depression, and anxiety [9,10,11]. For instance, Hermawati et al. (2018) conducted an empirical study on the use of digital media by infants under 2 years old. They found that children who used more than 3 h of screen time per day had symptoms such as attention problems and hyperactivity [11]. Additionally, the most frequent symptom of IGD across all age groups (adolescents, adults, and the general population) seems to be depression. Furthermore, according to research on the psychiatric characteristics of frequent Internet users, there is a high prevalence of symptoms such as social anxiety, emotional issues, and cognitive deficits. [12,13,14]. Moreover, digital addiction affects children and adolescents psychologically and physically, e.g., vision loss, hearing impairment, and body obesity [15,16]. At the same time, it will also affect children’s sleeping and eating patterns, causing substantial damage to their health [17]. In addition, studies have demonstrated that addiction has specific effects on brain function. For example, Han et al. (2018) revealed that adolescent IGD was related to the altered position of some prefrontal–striatal circuits [18]. However, despite these various and severe impairments, DA is treated to date unsystematically, and review studies investigating intervention in individuals with DA, especially in children and adolescents, are rare.

### 1.3. COVID-19 Exacerbated the Digital Addiction Problem

The worldwide lockdowns during the COVID-19 pandemic have resulted in a sharp increase in the cases of digital addictions, especially in children and young adolescents. Most preschools, schools, and colleges have switched to home-based online teaching and learning, and hundreds of millions of children worldwide have also been transformed into “online learners”. Using India as an example, during the epidemic period, primary and middle school students used digital devices for an average of 2.11 h per day, with a 28.1% incidence rate of digital addiction [19]. However, neither these children nor their teachers were ready for such a sudden paradigm shift to online learning and teaching, theoretically and practically. Additionally, parents were not yet prepared for digital parenting at home. Therefore, these homeschooling children were really vulnerable to excessive or unrestricted use of digital devices, which caused DA accordingly. The negative consequences of Internet use may be amplified in the context of COVID-19. Due to increased access to technology and online media, a lack of other sources of entertainment, and a reliance on the Internet for communication with the outside world brought by the stay-at-home order, the prevalence and severity of DA among children and adolescent populations have increased during this period. For instance, Liu et al. (2021) surveyed 1121 medical students in China during the COVID-19 pandemic and found that mobile phone addiction was highly correlated with their food and sugar intake [20].

Additionally, addiction is known to have a greater risk of harming a child’s or adolescent’s development because they have not yet fully developed their identities and have few coping mechanisms than adults. This risk is higher the younger addiction starts [7]. Children and adolescents have less self-control than adults, especially in social media and online games, and relatively quickly suffer from Internet addiction. According to the social cognitive theory, the learning and development of individuals in childhood and adolescence are influenced by the social environment, from the micro level—communication with parents and peers—to the macro level—the changes of the whole society [21]. Therefore, the emergence of digital devices and online learning replaced traditional parent–child or peer interactions. Coupled with the decline in outdoor activities during the COVID-19 pandemic, online games and social media are the preferred methods of alleviating boredom for children and adolescents. In addition, childhood and adolescence are critical periods in brain development [22], and digital addiction may cause functional damage to the brain [23,24]. The damage will be permanent and irreversible if not diagnosed and treated in such critical periods. Therefore, it is significant to implement early prevention or timely intervention in children and adolescents to prevent or cure digital addiction. The treatment cost and prognosis are better and more effective than later interventions in middle school and college. Currently, there have been some systematic reviews of prevention and intervention studies targeting Internet addiction among university students and adults, such as Zhang et al. (2022), Xu et al. (2021), Vondráčková and Gabrhelik (2016), and Kuss and Lopez-Fernandez, (2016) [25,26,27,28]. Still, none of them have reviewed those studies focusing on children and adolescents (ages 0–18), especially during the past 5 years (2018–2022). Without synthesized evidence, health officers, scholars, practitioners, educators, and psychiatrists, might have difficulties identifying non-invasive and effective intervention programs for digital addiction for children and adolescents. To fill this research gap, this present study examined the existing DA interventions among children and adolescents, reviewing and comparing their effectiveness to inform interventions that could appropriately reduce health risks and improve digital health. The following research problem guided this study:

What are the major intervention programs for digital addiction in children (ages 0–18)?

## 2. Methods

This present study employed the scoping review methodology to synopsize research findings for related stakeholders, find research gaps, and identify fields for further research [29,30]. This method allows this present study to investigate the breadth and depth of existing research on treating digital addiction, digital overuse (including internet television, tablet, smartphone, and video game addiction or overuse) intervention among children and adolescents, and identifying areas for additional research and research gaps. In this study, potential sources were specifically sought out, noted, gathered, and assessed for relevance to this study’s goals before being mapped to the major themes and ideas that underlie the research questions. Following Arksey and O’Malley’s (2005) [21] scoping review framework, five steps were taken: defining the research questions, identifying relevant studies, choosing studies, charting the data, and compiling, summarizing, and reporting the findings.

Step 1: defining the research questions.

The research questions were proposed to guide this review: (1) What are the major types of DA intervention? (2) What is the main content of the DA interventions for children and adolescents? (3) How effective are these DA interventions? (4) What are the research gaps in this area of study?

Step 2: identifying relevant studies.

This review was conducted by searching Science Direct, Web of Science, and Google Scholar for relevant research in December 2022. The last search date was 18 December 2022. The goal of the literature search was to locate every study on “digital addiction interventions in children and adolescents” published during 2018–2022. To find and extract the appropriate literature from the databases, three different sets of terms with two Boolean operators (AND and OR) were used: (internet OR digital OR TV OR smartphone OR tablet OR video game OR internet game) AND (preschoolers OR children OR young children OR adolescent OR youth OR teenagers) AND (intervention OR treatment OR therapy). The development of search terms involved extensive piloting.

Step 3: studies selection.

Only peer-reviewed, full-text, English journal articles that matched this review’s objectives were included. Following were the criteria for inclusion: (1) published during the period of 2018–2022; (2) evaluated a psychological intervention or treatment on digital (e.g., Internet, T.V., video game, smartphone, or tablet) addiction; (3) resulted from the report on digital addiction interventions covering young children and adolescents (ages 0–18); (4) original studies; and (5) the written language was English.

The articles were excluded if they were: (1) single case designs; (2) review or theoretical papers; (3) not available in English; and (4) not including the age range of children (under 18 years).

Following Moher et al. (2009) [31], the flow of information through the different phases of this present review is shown in Figure 1. After removing 2505 duplicate articles from the final search results, 2603 articles were left. The first and second authors reviewed and chose the articles independently using the inclusion criteria. The authors then went through full-text articles, screened them for inclusion, and extracted data from them. Fifty-eight studies were disqualified by title and abstract out of the total included for full-text review. In addition, 19 of the 40 full-text studies whose eligibility was evaluated were disqualified. The authors debated the studies’ eligibility until they reached a 100% agreement. Finally, 17 psychological intervention studies of digital addiction among children and adolescents were qualified for review.

Step 4: charting the data.

Finally, this present scoping review concluded 17 articles that included cognitive-behavioral therapies (CBT), CBT-based interventions, family therapies, and digital-based interventions. Most of the research came from South Korea (*n* = 4), Germany (*n* = 4), and China (*n* = 3). The characteristics of the included studies are summarized in Table 1, and intervention details are concluded in Appendix A.

Step 5: collating, summarizing, and reporting results.

This review summarized the sources in five types of intervention, including CBT, CBT-based therapy, family therapy, digital-based intervention, and other interventions. The table was adapted from Xu et al. (2021) [26], which is the most recent review of Internet addiction and IGD interventions. Similarly, this review also focused on the intervention among children and adolescents and added digital-based intervention as a part. Additionally, to highlight the need for caution when interpreting the reported results, the limitations of each study were listed in the table.

## 3. Results

This present study included those studies published in peer-reviewed journals during the 5 years 2018–2022 to examine the latest advances. A total of 17 studies were classified into 5 categories: CBT, CBT-based intervention, family therapy, digital-based intervention, and other interventions. Overall, the proposed countermeasures effectively reduce digital addiction in children and adolescents.

### 3.1. Cognitive-Behavioral Therapy Intervention

As the most evidence-based, transdiagnostic psychotherapy approach [47], CBT is effective in treating substance abuse, gambling, affective, and eating disorders. CBTs are based on the cognitive–behavioral model, which holds that thoughts determine feelings, and thus changing one’s thoughts can help with behavioral change [48,49]. Therefore, CBT has been widely used to treat problematic Internet use, and cognitive–behavioral models for understanding the development and maintenance of IA have been proposed [50].

CBT is broken down into three stages. The first stage is behavior modification, which gradually reduces the time people spend online and creates a healthy Internet use schedule. The therapist assists the individual in developing a schedule that includes non-Internet-related activities to reduce pathological use [51]. This phase aims to help students manage online and offline time [52]. The second stage is cognitive reconstruction and rationalizations that justify excessive Internet use. This stage seeks to identify and reverse the triggers for overuse and correct the cognitive conditioning that drives the individual to initiate Internet use. The third stage concentrates on the individuals’ functional issues related to their Internet use, both individually and professionally, to aid in identifying and treating coexisting issues that may have contributed to the development of problematic Internet use. The goals of this phase are to maintain recovery and relapse prevention. Overall, by focusing on these three main goals—reducing usage hours, improving functioning in essential areas of life, and minimizing exposure to content and problematic online operations—CBT and other CBT-based psychological interventions may help lessen the severity of Internet addiction [53].

Four studies jointly found that CBT might be the most effective treatment for decreasing Internet use and increasing self-awareness in users. First, following the completion of a brief CBT program, 54 adolescents aged 9 to 19 (16.7% of whom were Internet addicts) showed a significant decrease in Internet usage times as well as the emotional and physical side effects of use. [32]. This school-based prevention program employs cognitive behavioral techniques such as psychoeducation, cognitive restructuring (identifying and changing dysfunctional beliefs), and life skills (problem-solving, behavior modification, and emotion regulation). Each module addressed the potential risk factor in developing and maintaining IUD. The finding suggests a brief four-session intervention can have a medium-to-large impact over a year. [32]. Second, Kim et al. (2018) found that an 8-session (1-month) group CBT program was successful in reducing the symptoms of concurrent depression and anxiety, and the effectiveness of the program was thought to last for up to 1 month. [33]. The eight sessions included building intimacy through self-introduction and games, understanding the advantages and disadvantages of using the Internet/looking back at Internet habits, identifying the patterns of and reasons for problematic Internet use, working on self-reviews about Internet use, monitoring Internet usage, setting target time for Internet usage, setting practical application of time reduction on the Internet, and setting goals to prevent a recurrence. Third, Pakpour et al. (2022) delivered eight consecutive sessions of an app-based intervention based on the transtheoretical model and CBT in two months for adolescents. This study focused on three domains, i.e., stages of change, decisional balance, and self-efficacy. The findings suggested that this therapy could be used with other treatments for adolescents with IGD. [34]. Therefore, CBT may be effectively treated Internet addiction in the short term among adolescents.

In particular, cognitive–behavioral treatments have the potential to improve self-control, depression, and anxiety and reduce symptoms of Internet addiction in the adolescent population. According to research [35], which applied cognitive–behavioral therapy for addictive behaviors among seventh to nineth graders, the experimental group had higher levels of self-control than the control group, indicating preliminary efficacy in reducing Internet addiction. Kim et al. (2018) also revealed that participants’ addiction, depression, and anxiety scores were significantly lower than after group CBT intervention [33]. Furthermore, neuroimaging studies on male IGD participants provided evidence for the positive effects of CBT intervention among adolescents, demonstrating that the symptoms of IGD may be reduced with CBT by regulating the abnormal low-frequency fluctuations in prefrontal–striatal regions in IGD subjects. The degree of functional connectivity changes in IGD subjects was positively correlated with the scale of the Internet addiction scores changes. Meanwhile, after CBT, the weekly gaming time was significantly reduced, and the performance of the Internet addiction score and behavioral inhibition function was improved considerably [18]. As a result, CBT programs for Internet addiction in adolescents are thought to effectively reduce Internet addictions and their related symptoms in general.

### 3.2. CBT-Based Intervention

CBT-based intervention is a type of compound psychotherapy that combines CBT with other treatments, such as pharmacology, psychotherapy, and family counseling, either concurrently or sequentially. This compound approach may assist in addressing complex concerns. Combining CBT and other therapies with different treatment targets and mechanisms may act synergistically. For instance, Kim et al. (2018) conducted a CBT–music therapy program among Internet overuse adolescents, and the results showed that the Internet addiction, depression, and anxiety scores of participants were lower than after a one-month follow-up group intervention [33]. The Internet addiction scores remained low at the one-month follow-up assessment, and the depression and anxiety scores were even lower than immediately after the program. Similarly, CBT combined with music therapy was successful in treating smartphone and Internet addiction in adolescents between the ages of 11 and 16 and reduced impulsivity, anxiety, and other symptoms [36]. The intervention included eight weeks of MT combined with CBT Home Daily Journal writing in adolescents. The game and immersion MT program manual, which was approved by the National Association of Korean Music Therapists, served as the basis for the music therapy program.

Kochuchakkalackal and Reyes (2019) underscored the importance of integrating CBT and mindfulness theories in cognitive restructuring to alleviate IGD symptoms in adolescents and improve their psychological health through the Acceptance and Cognitive Restructuring Intervention Program (ACRIP) program [37]. The ACRIP program integrates the theories of the Cognitive–Behavioral Model of Pathologic Internet Use and Mindfulness. The modules had a logical progression that began with the introduction of the ACRIP and its goals to the participants, encouraged them to be open and accept their thoughts and emotions, helped them to let go of their negative outlook/accept who they are/rekindle broken relationships, and gave them the tools they needed to manage their thoughts and emotions with a strong will to accomplish their goals [37]. In this study, 10 adolescents received an 8-module Acceptance and Cognitive Restructuring Intervention Program (ACRIP) program and showed a significant difference post-test. The findings suggested that ACRIP might be an effective intervention program for reducing IGD in adolescents and improving their overall psychological well-being. In addition, this program may also be used to prevent the development of IGD and to break the cycle. As a result, CBT has the potential to treat adolescents with IA or IGD and may also help with symptoms of depression and anxiety.

Pornnoppadol et al. (2020) recruited 104 adolescent children in 1 of their experimental groups through a 7-day Siriraj Therapeutic Residential Camp (S-TRC), which included 10 sessions of group CBT, 4 media literacy sessions, and 2 workshops [38]. The objectives of S-TRC are to encourage life–game balance and healthy gaming behavior. They found that the symptoms of Internet addiction had improved after the intervention. In addition, participants in this camp were immersed in outdoor and indoor activities designed to boost their self-esteem, social skills, problem-solving abilities, self-awareness, and self-motivation [38]. Moreover, Lindenberg et al. (2022) recruited 167 Internet-addicted adolescents and implemented the PROTECT intervention. They found the theory-driven, school-based, manualized, CBT-based preventive group intervention successfully decreased symptoms of gaming disorder and an unspecified Internet use disorder over a year [39]. Furthermore, Yang and colleagues (2018) combined and used self-efficacy and self-regulation intervention among 13–15-year-old adolescents and discovered that Internet addiction and time spent online significantly decreased in the intervention group compared to the control group, while self-control and self-efficacy significantly increased [40].

There are also some effective interventions to reduce the prevalence of Internet addiction among adolescents and addiction severity through learning healthy Internet knowledge. For example, the Healthy Internet Use Program is a three-month school-based training intervention. The Program is based on the social cognitive theory, which describes human behavior in terms of individual characteristics and interactions between behavior and environment. As it is the most popular theoretical framework for behavior-based programs intended to change people’s behaviors, the program could be used to treat Internet addiction [21]. Uysal and Balci (2018) discovered that the difference between the baseline was the same before the program implementation, and the scores after the third month significantly changed after the program. Additionally, after nine months, there was a highly significant difference in the scores used to measure Internet addiction between the intervention group and the control group, indicating that the Healthy Internet Use Program helps reduce the prevalence of Internet addiction among teenagers [21]. Similarly, Khoshgoftar et al. (2019) conducted an educational intervention based on the Health Belief Model (HBM). According to this model, people’s actions reflect their beliefs about their perceived susceptibility, severity, benefit, barrier, cues to action, and self-efficacy. Active learning techniques, including focus group discussions, movies, role playing, question and answer sessions, and lectures, were employed to deliver the lessons. Results showed that the intervention could reduce and prevent female students’ addiction to mobile phones [41]. In particular, two months after the intervention, the mean score of mobile phone addiction among the students in the intervention group decreased, whereas the score for the control group increased. In addition, constructs of the HBM, except for perceived barriers to reducing mobile phone use and perceived benefits of mobile phone use, significantly increased in the intervention group compared to the control group. However, the findings could not be generalized because it was a quasi-experimental study.

### 3.3. Family Intervention

Family is significant to the growth and development of children and adolescents, providing emotional connections and behavioral constraints [54]. Parenting styles, familial relationships, and familial functioning are all vital considerations for adolescent behaviors such as IA. Adolescents who do not receive adequate attention and support from their parents are more likely to develop psychological problems, which may lead to them using the Internet excessively to escape negative feelings about their home situations. A lower risk of developing IA was linked to good family functioning [55]; thus, family factors may be crucial IA intervention targets.

In contrast to other interventions, family therapy is a series of interventions intended to enhance family dynamics rather than specifically address addictive behaviors. To treat internet gaming disorder and its comorbid disorders and symptoms, as well as to improve intra- and interpersonal skills and family relationships, Torres-Rodrguez et al. (2018) developed the PIPATIC (Programa Individualizado Psicoterapéutico para la Addicción a las Tecnologias de la Información y la Comunicación) program [42]. They compared the efficacy of PIPATIC intervention and standard CBT to internet gaming disorder among 12–18 adolescents. They found significant differences in comorbid conditions, intrapersonal and interpersonal abilities, and family relationships between the pre-test and post-test results. Both groups also noticed a considerable decrease in Internet gaming disorder symptoms, though the PIPATIC group saw more significant improvements in the other variables under study.

In addition, parents’ thoughts and attitudes toward digital devices also have a significant influence on their kids, especially adolescents. Pornnoppadol et al. (2020) found that an 8-week program named Parent Management Training for Game Addiction (PMT-G) was an effective psychosocial intervention for internet gaming disorder. This program outperformed basic psychoeducation in enhancing parental understanding of the causes of game addiction, improving parenting skills in dealing with the problems, and reducing family conflict (control group) [38]. The effectiveness of PMT-G was primarily due to an improvement in the parent–child relationship. Families may communicate more empathetically by educating parents and influencing their attitudes toward gaming. Developing discipline may lead to more organized and regular reinforcement at home. Learning new communication patterns, such as validation, reflective statements, and assertive communication, fosters a calmer home environment that is more receptive to change. Finally, adolescents may be more willing to change if they see their parents attempting to change. Therefore, this review found that some researchers also adopted family interventions.

### 3.4. Digital-Based Intervention

The development of new technologies frequently brings new challenges for maintaining, enhancing, and recovering mental health and well-being, but they also open up the possibility of new, successful interventions. Digital-based intervention is one of the most promising interventions in digital (internet, smartphone, computer, tablets, etc.) addictions. “Digital-based” refers to those cutting-edge solutions that use digital platforms to provide assessment, prevention, and intervention. Approaches such as web-based tools, mobile applications, wearables, and virtual reality platforms effectively evaluate and treat a wide range of mental health disorders, particularly depression, anxiety, and substance and behavioral addiction [56]. For example, Chau (2019) implemented a Wise IT-use program (combined online training and offline workshop) among 7–13-year-old adolescent children and found that in 2-month follow-up assessments, IGD symptoms and the percentage of students at risk for the condition decreased after the program [43]. This program was designed based on three fundamental components: assimilation, interaction, and reflection. First, the learning principles and materials on safe and healthy online behaviors were created to be “injected” or assimilated into games for the assimilation component so that players learn while immersed in the games. Second, through the play-based activities incorporated into the learning sessions, students were encouraged to interact with and learn from one another as part of the interaction component. Third, the program made sure that students would be led by facilitators through a variety of activities that required them to reflect on their prior online experiences for the reflection part of the curriculum. Deep learning and knowledge retention are made possible by such engagement and reflection.

Moreover, with the development of mobile phones, applications have been recruited to intervene in digital addiction. For example, Lazarinis et al. (2019) created a mobile application to popularize the knowledge of Internet addiction to 10-year-old students through storytelling [44]. Each short story depicts a situation encountered by children while surfing the Internet. The tool aims to make students reflect on their online activities and change their attitudes by using the emotions of the virtual characters and the presence of specific visual clues. The application’s architecture allows for the integration of new stories that address fresh subjects or comparable problems from various angles. Each tale is divided into the following sections: a situation that a child must deal with, along with some visual cues of the problematic conditions; a situation in which the child may be in a better or worse state than before, with the problem being obvious; and finally, some images or textual cues that offer some solutions. Although this study is an introduction to storytelling application, it demonstrates that digital-based technologies can be applied to children’s Internet addiction intervention. In another study, Pakpour et al. (2022) developed an application maned HAPPYTEEN, which is a two-month app-based intervention based on the transtheoretical model and CBT. This study demonstrated that the application could be employed as an adjunctive treatment for adolescents with IGD [34]. It is worth noting that the application intervention has been personalized based on the situation in the process.

### 3.5. Other Interventions

Aside from the psychological interventions mentioned above, physical activity intervention is an important intervention method that positively impacts individual cognition, emotion, and physical fitness. Tseng et al. (2022) conducted a 12-week strategic physical activity intervention among school-aged children, demonstrating that a well-planned physical activity program could be a practical and successful behavioral strategy to improve motor and cognitive skills in digital-addicted children [45]. In addition, Gong et al. (2022) evaluated the effect of narrative therapy combined with 8 reps of Pilates exercises on 42 adolescents with Internet addiction [46]. Adolescents in the intervention group clearly showed less Internet addiction than those in the control group, with a statistically significant difference between before and after the intervention. The experiment group, in particular, had significantly higher scores for mental health and significantly lower scores for anxiety, depression, social dysfunction, and loss of interest than the control group (no intervention) after the intervention. Through pre-post comparison, the intervention group also demonstrated a statistically significant improvement in the score for positive affect.

## 4. Discussion

### 4.1. Five Types of DA Intervention Programs

This literature review found that the studies collected in this study could be classified into five categories: CBT, CBT-based intervention, family therapy, digital-based intervention, and others. First, CBT and CBT-based psychotherapies were the most commonly used psychotherapy for DA. It is superior to other addiction disorder interventions (e.g., hospitalization or drug substitution therapy), is less invasive [57], and can be effective in the short term. In addition, CBT is the most effective intervention for reducing Internet use and improving self-perception in chronic users, producing improvements in a short period that are maintained for at least six months [33,34]. This type of therapy works to reframe negative thoughts, assisting adolescents in giving a new sense to routine and problematic behaviors and motivating the creation of a more adaptive, goal-oriented routine [35] to improve symptoms of depression and anxiety. Second, family therapy aims to improve parent–adolescent communication and relationships and redirect adolescents’ psychological needs from the Internet and toward interactions and relationships with family members. Children and adolescents’ thoughts are immature during childhood and adolescence, making them more susceptible to the influence of their surroundings. They also have less self-control and are more prone to digital addiction or overuse. As a result, it is critical to establish a positive attitude toward and use of the Internet through CBT and family support. Third, digital-based intervention was also widely adopted because the COVID-19 pandemic had increased online courses and younger users. More importantly, research has shown that the COVID-19 pandemic has increased young children’s time using digital devices and contributed to developing a digital addiction. Therefore, traditional modalities (such as face-to-face therapies) may no longer be able to help prevent, treat, monitor, and manage IGD due to restrictive measures, posing a barrier to therapeutic intervention. Digital technology, more than ever, is likely to play an essential and irreplaceable role in addressing the digital addictions that arose during and before the pandemic. Some of the currently available and recommended solutions include: (1) using websites or apps to spread awareness of the potential risks of gaming during the lockdown period, thereby assisting children and young people in developing good digital device use knowledge; (2) providing health guidelines for integrating digital devices use into the learning and life of individuals (3) proposing types of digital intervention games that can promote mental and physical health, thereby enabling children to get rid of digital addiction in video games and achieve “fighting poison with poison” [58].

### 4.2. Digital Intervention to Cure Digital Addiction

Psychological interventions have significantly changed with the widespread availability of digital technologies such as computers, the Internet, mobile devices (smartphones), and mobile applications. Digital-based interventions such as online intervention, virtual reality, and applications have made intervention methods more similar to offline interventions while remaining cost-effective and often easier to deliver than face-to-face interventions [34,44]. Consequently, digital intervention is unaffected by other factors and can be carried out more effectively. Online courses, particularly during the COVID-19 pandemic, could save manpower and material resources while being more accessible and efficient. When the special period passes, online and offline interventions consolidate knowledge learned in online training modules and promote real-life knowledge application [43].

The existing evidence suggests that further research into digital-based interventions for digital addiction is merited. First, numerous studies have shown that digital interventions are effective, particularly regarding adolescents’ digital device cognition and related symptoms. Second, one of the primary benefits of this intervention is its accessibility. Today, more than 50% of people on the planet have access to the Internet. In some countries, more than 95% of the population uses the Internet [58], demonstrating the viability of digital-based intervention. This is especially important for developing or low-income countries, where the burden of mental health disease is disproportionately high due to underdeveloped healthcare systems, a lack of access to intervention, and a scarcity of mental health professionals [59].

### 4.3. Why Digital Intervention Is More Promising than Others

Digital intervention is more promising than other strategies, as it is “fighting poison with poison” [58] and has three major advantages. First, it has better availability. Where other methods were unavailable during the COVID-19 lockdowns, digital-based interventions provided a unique and promising chance to increase the psychological health of individuals, no matter their wealth, race, gender, or age. As a result, they could reach a wider audience than traditional therapies. Second, it has better accessibility. Digital intervention is easy to use and convenient to access, regardless of the time, location, or number of participants. Third, digital interventions could be better individualized, including adapting treatment programs to specific populations and individual needs, anonymity, and, most importantly, lower costs [34]. Therefore, they are viewed as equally superior, less expensive alternatives [60]. For these reasons, digital intervention has been identified as a “critical component” of the future of mental health care [61]. As the population of DA children and adolescents grows, as does the demand for prevention and intervention, traditional intervention limitations should be overcome, and digital-based interventions developed.

However, digital intervention has some limitations in treating digital addiction. Because of the early stage of research on digital-based interventions, and the lack of extensive sample size studies, the results of existing studies can only serve as opening evidence of the effectiveness of digital-based interventions for DA. In addition, barriers to implementing these tools are rarely discussed, and there is little understanding of how best to use these solutions for individuals and integrate them into an all-in-one solution. Furthermore, similar to standard care, studies of digital-based interventions lack uniform assessments; thus, future research is required to confirm the efficiency of digital-based interventions.

Nevertheless, this review has highlighted the potential and significance of digital intervention. In the future, digital intervention should be “fighting poison with poison,” utilizing technology to treat digital addiction. As shown in Figure 2, individuals who require intervention should be screened out of the evaluation of digital addiction, and customized interventions should be carried out based on the various stages of individuals. In addition, the measurement should be performed while the intervention is being implemented, such as pre-post assessment. If the objective of the intervention is achieved, follow-up assessments are required; if the aim is not achieved, a new turn with fresh assessment and intervention is needed for the addict. An integrated evaluation and intervention system can thus be established. Furthermore, the system should be easy to use and accessible across multiple platforms (websites, smartphones, tablets, and so on). Moreover, it could be combined with various intervention modalities, allowing digital addicts to provide customized interventions. Finally, a better understanding of the mechanisms and underlying models of DA will be deepened through the screening of early digital addiction and follow-up investigations.

## 5. Conclusions

This literature review has identified and analyzed the 21 studies of DA intervention published between 2018–2022. The finding revealed that, first, most interventions for digital addiction in children and adolescents were cognitive–behavioral therapies (CBT) or CBT-based interventions, which could improve anxiety, depression, and related symptoms of digital addiction. Second, rather than directly targeting addictive behaviors, some family-based interventions aim to strengthen family functions and relationships. Finally, digital-based interventions, such as website-based, application-based, and virtual reality interventions, are promising in adolescent digital addiction interventions. Additionally, these studies shared the same limitations: small sample sizes, short intervention durations, no control group, and nonrandomized assignments.

However, this review has some limitations. First, this review only included those English papers; thus, it might have overlooked critical findings published in other languages, potentially biasing the review result. Second, the settings of different interventions, such as online or offline interventions, individual or group interventions, etc., also influence the effect. This review did not make further distinctions between different intervention settings, as it would result in correspondingly small numbers of studies for which reliable conclusions can be drawn. Follow-up review studies could delve deeper into the effects of various interventions. Finally, as this present study is based on a review of the existing literature, which contains few interventions for DA in young children, the results might be influenced by this publication bias, and the findings should be regarded as preliminary. Future studies of young children are needed to confirm this.

Nevertheless, this review and its findings might have some implications for future studies. First, it was discovered that interventions effectively reduced DA symptoms, depression, and anxiety, but these findings were mostly restricted to adolescents. With the younger age of DA, more intervention measures for children’s DA should be explored, and long-term studies with a large sample size are required to determine whether the intervention has long-term effects. Secondly, more preventive research is needed than intervention studies targeting individuals with DA. Establishing proper DA literacy through educational initiatives is critical to preventing, detecting, and treating DA. This proactive approach could help individuals understand and manage their digital device usage before it develops into problematic use requiring intervention. Thirdly, our review reveals that most current intervention and prevention approaches involve face-to-face interactions, either individually or in groups. This method has high requirements on the quality and quantity of therapists, there are certain limitations in the distribution of regions and services, and there are particular challenges in providing personalized, customized solutions. Therefore, digital-based countermeasures and digital interventions such as online courses may provide DA intervention to many needy people. In addition, digital-based countermeasures help reduce digital addiction because of their high accessibility, digital surveillance, and automated real-time intervention. Moreover, in the design of the digital intervention, the diagnosis, intervention, and assessment should be integrated into an integrated system, which helps to maximize intervention convenience and practicability. Fourthly, future research should focus on individualized interventions for DA and comorbid mental disorders with DA, such as attention deficit hyperactivity disorder, depression, mood disorders, impulse control disorders, and social avoidance disorders. Although a causal relationship between DA and comorbid mental disorders is difficult to confirm, comorbid diseases can be evaluated as risk factors for DA so that individualized intervention can be carried out to achieve the desired effect. Last, DA intervention evaluation should include neuroimaging tools, as DA impacts human brain function. As a result, using behavioral and neuroimaging tools in conjunction with subjective and objective evaluation methods may provide biological evidence of the intervention.

## Figures and Tables

**Figure 1 ijerph-20-04777-f001:**
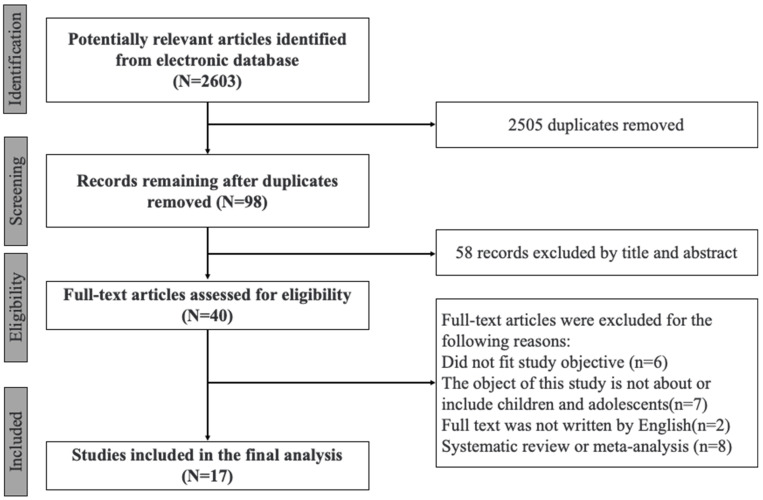
The selection of study included in this scoping review.

**Figure 2 ijerph-20-04777-f002:**
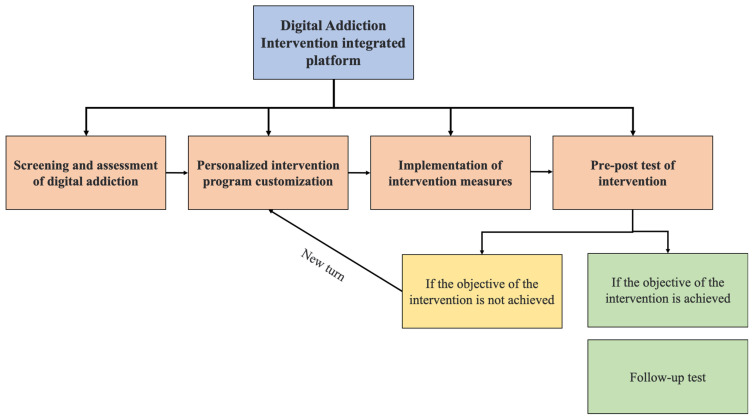
Schematic diagram of the Digital Addiction-integrated intervention platform. Blue represents the first level; orange represents the second level, divided into screening and assessment, intervention customization, intervention implementation, and pre-and post-testing; yellow and green represents the third level, green represents the achievement of the intervention objective, and re-evaluation the sustained effect of the intervention; yellow represents the failure to achieve the intervention objective, and the need for a new round of intervention.

**Table 1 ijerph-20-04777-t001:** Digital Addiction Intervention for Children and Adolescents.

Author /Year	Country	Addiction Type	Sample Size	Age (Years Old)	Participants’ Details	Assessment	Intervention	Control Group	Result	Limitations
Han et al., 2018 [18]	China	IGD	*n* = 26	16.81 ± 0.75	meet the standards of the Diagnostic Questionnaire for Internet Addiction test	The Chen Internet Addiction Scale (CIAS); The Self-rating Anxiety Scale (SAS); The Self-rating Depression Scale (SDS); and The Barratt Impulsiveness Scale-11 (BIS-11)	CBT	/	CBT could regulate the abnormal low-frequency fluctuations in prefrontal–striatal regions in IGD subjects and improve IGD-related symptoms.	lack of control group; only males
Szász-Janocha et al., 2020 [32]	Germany	IUD	*n* = 54	13.48 ± 1.72	self-reported or parent-reported excessive gaming or Internet use and who reported subjective psychological strain (self-selection)	The German Video Game Dependency Scale (CSAS); The compulsive internet use scale (CIUS); The Strengths and Difficulties Questionnaire; The Children’s Depression Inventory; The Social Interaction Anxiety Scale; The Fear Survey Schedule for Children-Revised; The German Questionnaire for Procrastination; The German Questionnaire for Assessment of Emotion; Regulation in Children and Adolescents; The German Student Assessment List for Social and Learning Behavior; The German General Self-Efficacy Scale	the PROTECT + program	/	Patients showed a significant reduction in IUD symptom severity at the 12-month follow up. Secondary outcome variables showed a significant reduction in self-reported depression, social anxiety, performance anxiety, and school anxiety, as well as in parental-reported general psychopathology	lack of control group; did not use objective inclusion criteria for study participation
Kim et al., 2018 [33]	South Korea	IA	*n* = 17	12–17	screened for psychiatric disorders with the Mini International Neuropsychiatric Interview for Children and Adolescents (MINI-KID)	Korean Internet addiction self-report test (K-scale); Young’s Diagnostic Questionnaire (YDQ); Young’s Internet Addiction Scale (IAS); Children’s Depression Inventory (CDI); State-Trait Anxiety Inventory (STAI)	group CBT	/	After the program, the IAS, CDI, and SAI scores were lower than before. At the one-month follow-up assessment, the IAS scores remained low, and the CDI and SAI scores were even lower than after the program.	only implemented in two specific schools; no control group; self-report evaluation scales only; only voluntary participants with a willingness to decrease problematic Internet use participated in the study
Pakpour et al., 2022 [34]	Iran	IGD	*n* = 206	13–18	scored 32 or higher on the nine-item Internet Gaming Disorder Scale—Short Form (IGDS-SF9)	Internet Gaming Disorder (IGDS-SF9); The Depression, Anxiety and Stress Scale (DASS-21); Insomnia Severity Index (ISI); Stages of Change Questionnaire (SOCQ); Decisional Balance Scale; and Self-efficacy Scale.	based on the TTM and CBT application (named HAPPYTEEN)	a sleep hygiene intervention via the app	This application intervention could be used as an adjunct therapy for adolescents with IGD.	lack of detailed results of the pre-post comparison
Agbaria, 2022 [35]	Israel	IA	E: *n* = 80 C: *n* = 80	E: 13.45 ± 1.46; C: 13.91 ± 1.92	high scores on Young’s (1998) Internet Addiction Test questionnaire	Adolescence Self-Control Scale; Internet Addiction Test (IAT)	CBT	Weekly classroom conversation	Higher levels of self-control were reported among the experimental group but not the control group, which contributed to a reduction in scores on the questionnaire assessing Internet addiction in the experimental group.	intervention time in after school hours; self-reporting questionnaires may be biased by reporting error, social desirability, or lack of insight into one’s behavior; did not examine individual differences, such as an adolescent’s level of impulsivity or executive functioning
Bong et al., 2021 [36]	South Korea	Smartphone and Internet Addiction	E: *n* = 67 C: *n* = 71	10–16	classified as high-risk (>70) or problematic (>40) users based on Young Internet Addiction Scale (YIAT), or smartphone high-risk group based on Korean Smartphone Addiction Proneness Scale (SAPS)	Young Internet Addiction Scale (YIAT); Smartphone Addiction Proneness Scale (SAPS); State Anxiety Inventory for Children (SAIC); Trait Anxiety Inventory for Children (TAIC); Conners-Wells’ Adolescent Self-Report Scale-Short form (CASS(S)); Barratt Impulsiveness Scale-11	CBT + Music therapy	CBT	Combined MT and CBT improved the symptoms of smartphone/internet addiction, anxiety, and impulsivity in adolescents.	the smartphone/internet addiction scales used in this study differed from the diagnostic criteria of internet gaming disorder in DSM-5 or gaming disorder in ICD-11; no follow up interventions
Kochuchakkalackal and Reyes, 2019 [37]	Philippines	IGD	*n* = 10	14–18	have a high score in the IGD and a low score in Ryff’s Psychological Well-Being scales; play more than 30 h per week and have manifested at least five of nine symptoms of IGD in the last 12 months.	Personal Data Sheet/Demographic Information Form (DIF); Internet Gaming Disorder (IGD) Scale; Ryff’s Psychological Well-Being (PWB) Scale	The Acceptance and Cognitive Restructuring Intervention Program (ACRIP) program	/	The results show a decrease in the adolescents’ IGD level and an increase in PWB level after the feasibility test, as indicated by the mean score of the post-test; and there is a significant difference in the pre-test and post-test scores of both IGD.	a pilot study; the sample is small and single
Pornnoppadol et al., 2020 [38]	Thailand	IGD	E1: *n* = 24; E2: *n* = 24; E3: *n* = 26; C: *n* = 30	13–17	adolescents and/or parents of adolescents aged 13–17 years with a total score meeting the cutoff for problematic online gaming in the Game Addiction Screening Test (GAST)–Parent version	Game Addiction Screening Test (GAST); Game Addiction Quality of Life Scale (GAME-Q); Game Addiction Protection Scale (GAME-P); Pediatric Symptom Checklist-17 (PSC-17) Thai version	E1:7-day Siriraj Therapeutic Residential Camp (S-TRC) E2: Parent Management Training for Game Addiction (PMT-G) E3: combined S-TRC and PMT-G	basic psychoeducation	All experimental groups showed improvement over the control group. The percentage of adolescents who remained in the addicted or probably addicted groups was less than 50% in the S-TRC, PMT-G, and combined groups.	family assessment was not conducted at recruitment; a quasi-experimental study (intervention and control only, without randomization); no data on potentially confounding variables (e.g., family dynamics)
Lindenberg et al., 2022 [39]	Germany	IUD	*n* = 422	15.11 ± 2.01	screened for risk before study enrollment using the German version of the Compulsive Internet Use Scale (CIUS) and chose a CIUS score of 20 as the cutoff criterion, thus, included participants at moderate risk and high risk. subgroup: CSAS: Computerspielabhängigkeitsskala, a modified German video game dependency scale DSM-5	The German Procrastination Questionnaire (APROF); The Strengths and Difficulties Questionnaire (SDQ); The German Depression Inventory for Children and Adolescents (DIKJ); The Social Interaction Anxiety Scale (SIAS); The German revision of the Fear Survey Schedule for Children (PHOKI); The Assessment of Emotion Regulation in Children and Adolescents (FEEL-KJ); The German Student Assessment List for Social and Learning Behavior (SSL); The German General Self-Efficacy Scale (SWE)	PROTECT (Professional Use of Technical Media) intervention	assessment only	The PROTECT intervention effectively reduced symptoms of gaming disorder and unspecified internet use disorder over 12 months. The intervention did not change incidence rates of gaming disorder or unspecified internet use disorder.	limit the generalizability of the findings; an underpowered sample for the incidence analyses assessment of incidence rates needs to improve; differences between schools remain open to speculation
Yang et al., 2018 [40]	South Korea	IA	E: *n* = 38; C: *n* = 56	13–15	categorized in the general internet user group or potentially addicted user group according to the Internet Addiction Proneness Scale	The Self-Control Scale; The Self-Efficacy Scale; The Internet Addiction Proneness Scale; an assessment of internet addiction (internet usage time).	Integrated and applied self-efficacy and self-regulation intervention strategies	no intervention	Self-control and self-efficacy significantly increased, and Internet addiction and time spent on the Internet significantly decreased in the intervention group compared with the control group.	not a randomized control trial but a quasi-experimental design; exogenous confounding variables not included; measured with self-report, there is a possibility of social desirability bias
Uysal and Balci, 2018 [21]	Turkey	IA	E: *n* = 41; C: *n* = 43	11–16	scored 90 or over on the Internet Addiction Scale (IAS)	The Internet Addiction Scale	Healthy Internet Use Program	no intervention	After the Healthy Internet Use Program, the difference between the baseline measurement and scores after the third month was significant. In addition, the difference in the measurement of the IAS scores between the intervention group and the control group after the ninth month was highly significant, suggesting that the Healthy Internet Use Program decreases the rate of Internet addiction among adolescents.	only implemented with the students of two schools and cannot be generalized; no exact criterion to identify Internet addiction.
Khoshgoftar et al., 2019 [41]	Iran	Mobile Phone Addiction	E: *n* = 56; C: *n* = 56	E: 14.62 ± 0.52 C: 14.66 ± 0.83	have a smartphone; interested in participation; parental consent	Persian version of mobile phone addiction scale (MPAI); Health Belief Model constructs questionnaire	The educational intervention: Health Brief Model (HBM)	the usual school curriculum	Educational intervention based on the HBM can prevent and decrease mobile phone addiction in female students.	not internet addiction sample; only a quasi-experimental study could not generalize the finding; only student intervention, lack of parent’s education
Torres-Rodríguez et al., 2018 [42]	Spain	IGD	E: *n* = 17 C: *n* = 17	12–18	endorsing at least five or more of the nine IGD criteria according to DSM-5; scoring 71 or more on Internet Gaming Disorder Test (IGD-20 Test)	The Clinical Global Impression Scale-Severity of Illness (CGI-SI); The Clinical Global Impression Scale-Global Improvement (CGI-GI); Global Assessment of Functioning (GAF) scale; The Working Alliance Theory of Change Inventory	PIPATIC (Programa Individualizado Psicoterapéutico para la Adicción a las Tecnologías de la information y la communication) intervention program	standard CBT	Relating to the interventions examined, significant differences were found in the pre-test and post-test on the following variables: comorbid disorders, intrapersonal and interpersonal abilities, and family relationships. In addition, both groups significantly reduced IGD symptoms, and the PIPATIC group showed higher significant improvements in the other variables examined.	Self-report measures; small sample size; no randomization of group assignment; lack of a control group without treatment; homogenous sample (Spanish male subjects only)
Chau et al., 2019 [43]	Hongkong	IGD	*n* = 248	10.16 ± 0.97	Typical primary student with parental consent	The self-report version of the Korean Internet Addiction Proneness Scale; Risky Online Behavior questionnaire; The Positive and Negative Affect Schedule for Children-Short Form; The Children’s Loneliness Questionnaire; The Social Anxiety Scale for Children, and Pre-post measurements of symptoms of IGD, frequency of risky online behavior, and rating of emotional well-being	Wise IT-use program	/	After the program(2-month follow up), IGD symptoms and the percentage of students who were at risk for the condition decreased	Homogenous sample: high school students residing in Hong Kong
Lazarinis et al., 2020 [44]	Greece	IA	*n* = 42	6–10	Typical developmental children with parental consent	/	Storytelling application	/	The application makes children reflect on their online actions through visual stories with contrasting elements.	an application introduction essay without an experiment
Tseng et al., 2022 [45]	China	IA	*n* = 10	10.45 ± 0.68	Identified using the Chen Internet Addiction Scale (CIAS)	Movement Assessment Battery for Children-Second Edition (MABC-2)	A strategic physical activity intervention	/	The post-test scores showed significant improvements in accuracy in the pure, mixed, and switch trials and the manual dexterity and total score of the MABC-2 compared with the pre-test scores.	small sample size; no control group; assessment should more detailed
Gong et al., 2022 [46]	China	IA	E: *n*= 21; C: *n*= 21	adolescents	Internet addiction adolescents	The 12-item General Health Questionnaire; the Self-rating Young’s Diagnostic Questionnaire of Internet Addiction, and the Positive Affect Subscale	Narrative therapy + Pilates exercise	no intervention	Compared to the control group, the intervention group had significantly lower scores for anxiety, depression, social dysfunction, loss of interest, and significantly higher scores for mental health. In addition, the intervention group demonstrated an apparent increase in the score for positive affect after the intervention.	other factors not included; a small sample without detailed characteristic

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
