# Peer review of "Digital Addiction Intervention for Children and Adolescents: A Scoping Review"

_ijerph, 2023, doi:10.3390/ijerph20064777_

Round 1

Reviewer 1 Report (Previous Reviewer 1)

The authors have provided a thoughtful and extensive edit of the manuscript that have improved it and the previous shortcomings.

Author Response

Thank you a bunch!

Reviewer 2 Report (Previous Reviewer 2)

I have re-read the resubmitted article and can see that the authors have made significant changes.

However, I would like to note the following:

In the introductory part, where they mention the existing scopings/literature reviews, they state that certain papers are not focused on children and adolescents, i.e. they do not include them in their analysis, which is a piece of wrong information - in the paper of Vondrackova and Gabrhelik the mentioned age group is also covered

In the analysis, they included a paper where only a small part of the sample is the group they stated as the target group - I am talking about this paper here: Dieris-Hirche, J.; Bottel, L.; Pape, M.; Te Wildt, B. T.; Wölfling, K.; Henningsen, P.; ... Herpertz, S. Effects of an online-based motivational intervention to reduce problematic internet use and promote treatment motivation in internet gaming disorder and internet use disorder (OMPRIS): study protocol for a randomised controlled trial. BMJ open, 2021, 11(8), e045840. doi:10.1136/bmjopen-2020-045840 – in the aforementioned paper, a significant proportion of the sample is older than 18 years of age, so the presentation of these results may seriously affect their further interpretation.

One can see that a lot of effort has been put into the additional revisions of this paper, but I suggest that the authors read the paper in detail, after making all the necessary revisions check whether they have written the correct statements, whether they have observed all the inclusion criteria and the like because these are the elements that can significantly affect the scientific value of the paper. Especially since there are a large number of papers in this field dealing with the same or a similar topic - so it is the quality, brevity, and validity of the results and their presentation according to the highest standards that can distinguish this paper from others.

Author Response

Reviewer 3 Report (Previous Reviewer 3)

The authors have satisfactorily addressed the queries. The paper is fit for publication as it stands.

Author Response

Thank you a bunch!

Round 2

Reviewer 2 Report (Previous Reviewer 2)

The manuscript has been sufficiently improved to warrant publication in IJERPH.

This manuscript is a resubmission of an earlier submission. The following is a list of the peer review reports and author responses from that submission.

Round 1

Reviewer 1 Report

The paper reviews literature on interventions against Digital Addiction for children and youth. It is a well-written and important paper that however could gain from some additional detail. I have given a few points of suggestions for the revision.

The rationale of the study needs to be more precise. There is a very great age-span in the study, going across childhood and adolescence, this needs to be more spelled-out why this is. See, for example the following paper for a call for a greater sensitivity regarding ages, developmental stages and contexts for studies on screen effects on children: Hassinger-Das, B., Brennan, S., Dore, R. A., Golinkoff, R. M., & Hirsh-Pasek, K. (2020). Children and Screens. Annual Review of Developmental Psychology, 2(1), 69–92. https://doi.org/10.1146/annurev-devpsych-060320-095612

I, however, think that the very broad age spectrum of the paper can be a strength of the study as it shows how wide the problem of DA is. If this can be more elaborated in the study it could give it more clinical appeal as well.

On p.3 it is stated that "In addition, childhood and adolescence are critical periods in brain development, and digital addiction may cause functional damage to the brain. If it is not diagnosed and treated in a critical period, the damage will be permanent and irreversible.”

While this may be true, such a claim requires references.

In the results section, the authors do well in presenting the literature. There, however, could be a greater detail to some of the presentation as we are presented with the names of several efficient CBT-based programs but no detail to what the programs entail. Just a few brief added sentences of what these educational and digital interventions consist of can make the review appealing for a broader audience besides specialists.

Author Response

  1. The rationale of the study needs to be more precise. There is a very great age-span in the study, going across childhood and adolescence, this needs to be more spelled-out why this is. See, for example the following paper for a call for a greater sensitivity regarding ages, developmental stages and contexts for studies on screen effects on children:Hassinger-Das, B., Brennan, S., Dore, R. A., Golinkoff, R. M., & Hirsh-Pasek, K. (2020). Children and Screens. Annual Review of Developmental Psychology, 2(1), 69–92. https://doi.org/10.1146/annurev-devpsych-060320-095612 I, however, think that the very broad age spectrum of the paper can be a strength of the study as it shows how wide the problem of DA is. If this can be more elaborated in the study it could give it more clinical appeal as well.

Response: Thank you so much for your constructive suggestion, which is very helpful and insightful. Yes, we agree that the broad age spectrum of this study might be a strength, indicating how wide the DA problem is. In this R1, we have taken your advice and elaborated more on this point in the introduction (pages 3, line 109-117, reproduced below).

  1. On p.3 it is stated that "In addition, childhood and adolescence are critical periods in brain development, and digital addiction may cause functional damage to the brain. If it is not diagnosed and treated in a critical period, the damage will be permanent and “ While this may be true, such a claim requires references.

Response: Thanks for your kind advice, which has been thoroughly followed in this R1 (see page 3, line 118).

  1. In the results section, the authors do well in presenting the literature. There, however, could be a greater detail to some of the presentation as we are presented with the names of several efficient CBT-based programs but no detail to what the programs entail. Just a few brief added sentences of what these educational and digital interventions consist of can make the review appealing for a broader audience besides specialists. 

Response: Thanks for your great suggestion. In this R1, we have elaborated more on the intervention programs both in the main text and the supplemental material.

Reviewer 2 Report

Dear Author(s),

I have read in detail your paper entitled "Digital Addiction Intervention for Children: A Scoping Review". The paper is written with a clear vocabulary, follows a logical sequence and the English language is understandable and clear for the reader. Considering that numerous studies indicate that young people - adolescents and students - are among the groups that are particularly at risk of developing digital/internet addiction, research in this area is of extraordinary importance.

This paper's main research problem is to examine the major intervention programs for digital addiction in children. Four research questions were stated: (1) What are the major types of DA intervention? (2) What is the major content of the DA interventions for children and adolescents? (3) How effective are these DA interventions? (4) What research gaps exist in this field of study?. To answer the research questions, the author(s) conducted a scoping review (following Arksey and O’Malley’s scoping review framework) on 21 selected studies.

In general, this paper is excellent, and as a reviewer, I would really commend the logical structure of the paper, readability, fluency, clarity of information, and the like. That's why my review will be shorter because I think the paper is really of high quality. With (really) minor revisions, which I suggest to the author(s), the paper will be suitable for publication.

Line 32: I would suggest a slightly different wording of this sentence where it says "...playing mobile phones..." - we use mobile phones, we play video games, and we use social networks (or we are active on social networks, for example).

Wherever "digital addiction" is mentioned, I would suggest that the entire paper focuses on "Internet addiction" because that is what is discussed in all the papers you analyzed. In other words, all the interventions you presented actually target the prevention/treatment of Internet addiction. Because the term "digital addiction" can suggest that you also deal with this part of excessive use of digital technology, which would include, for example, Playstation, smartwatches, or other gadgets. Regardless of how we approach it, we are really talking about Internet addiction as an umbrella term (specifically - video game addiction and social networking addiction). I would therefore suggest that this either be explained in more detail in the introduction of the paper or that the entire paper focuses on "Internet addiction".

Lines 108-109: It is important to emphasize that there may not have been papers available to the authors / or available in the English language. As it is stated now, it’s actually not entirely correct, considering that there are quite a number of systematic reviews/literature reviews related to the review of (preventive and treatment) interventions for internet addiction (below I am attaching some of the more recent reviews):

·        https://www.sciencedirect.com/science/article/abs/pii/S0165032722006735

·        https://www.sciencedirect.com/science/article/abs/pii/S1734114014000322

·        https://www.sciencedirect.com/science/article/abs/pii/S0272735813000020

·        https://akjournals.com/view/journals/2006/5/4/article-p568.xml

·        https://pubmed.ncbi.nlm.nih.gov/34087672/

·        https://www.ncbi.nlm.nih.gov/pmc/articles/PMC4804263/

Overall, the paper is really well written, and I believe its quality is already at a high level but can be enhanced by considering the preceding suggestions.

With respect,

Reviewer

Reviewer 3 Report

The present study is a scoping review for Digital Addiction intervention for children in the time span of 2018-2022. In general, I did not find this study to significantly contribute to the literature or the field. Following are my observations for the study:

1.      The title suggests that the author studied digital addiction intervention for children. However, the inclusion criteria for the studies consist of age groups from 0 – 18 years. This implies that the title is misleading as the scoping review focused on a wider age group and not specific to children.

2.      The authors studied the digital addiction intervention and internet gaming disorder simultaneously. However, lately, many researchers have done a systematic review of internet gaming disorder intervention and digital addiction intervention separately. So, I am skeptical whether this particular study throws any new light in this regard. Additionally, most of the selected studies in the review already come from Xu, et al (2021) and Zazac, Ginley & Chang (2020) studies.

3.      The study lacks justification for scoping review over systematic review. Besides, it lacks the standard practice of review reporting based on the PRISMA guidelines.

4.      There are some mistakes in citing the references in the paper. One such example is related to the Liu et al. (2021) study cited on pg 2, line 97. The authors stated that this study found that mobile phone addiction was highly correlated with their food and sugar intake. However, this reference was based on mobile phone use distraction on pedestrian reaction time.

5.      Similarly, some of the facts need to be double-checked. For instance, on pg 4, line 176 authors write that “the table is adapted from Xu et al (2021), which is the most recent review of IGD interventions”. However, this (Xu et al, 2021) study is the systematic review for internet addiction and not IGD solely.

6.      Results section: The selection of studies is confusing. One of the inclusion criteria is the age group between 0 -18 years. However, authors have included studies like, Gonzalez-Bueso et al. 2018 (age range 12 -21 years), Shin et al. (2018) (age range 13-21; 14-22 years)

7.      Similarly, other studies have been included that do not match the inclusion criteria mentioned in the manuscript. For instance: Lazarini et al (2020).

8.      Digital based interventions and other interventions are the salient and interesting points mentioned in the paper. However, a paucity of studies included and deviation from the inclusion and exclusion criteria have limited its scope.

In view of the aforementioned points, I believe that this study does not contribute significantly to the literature and unfortunately, I cannot recommend the article for publication.

References

Xu, L. X.; Wu, L. L.; Geng, X. M.; Wang, Z. L.; Guo, X. Y.; Song, K. R.; ... Potenza, M. N. (2021). A review of psychological interventions for internet addiction. Psychiatry Research, 302, 114016. doi: 10.1016/j.psychres.2021.114016

Zajac, K., Ginley, MK., & Chang, R. (2020). Treatments of internet gaming disorder: a systematic review of the evidence. Expert review of Neurotherapeutics, 20 (1), 85-93. doi:10.1080/14737175.2020.1671824

Liu, Y.; Alsaleh, R.; Sayed, T. (2021) Modeling the influence of mobile phone use distraction on pedestrian reaction times to green signals: A multilevel mixed-effects parametric survival model. Transportation research part F: traffic psychology and behaviour, 81, 115-129

González-Bueso, V.; Santamaría, J. J.; Fernández, D.; Merino, L.; Montero, E.; Jiménez-Murcia, S.; ... Ribas, J. (2022). Internet gaming disorder in adolescents: Personality, psychopathology and evaluation of a psychological intervention combined with parent psychoeducation. Frontiers in psychology, 9, 787. doi: 10.3389/fpsyg.2018.00787/full

Shin, Y. B.; Kim, J. J.; Kim, M. K.; Kyeong, S.; Jung, Y. H.; Eom, H.; Kim, E. (2018) Development of an effective virtual environment in eliciting craving in adolescents and young adults with internet gaming disorder. PLoS One, 13(4), e0195677. doi:10.1371/journal.pone.0195677

Lazarinis, F.; Alexandri, K.; Panagiotakopoulos, C.; Verykios, V. S. (2019) Sensitizing young children on internet addiction and online safety risks through storytelling in a mobile application. Education and Information Technologies. doi:10.1007/s10639-019- 68209952-w

Round 2

Reviewer 3 Report

I appreciate that the authors have incorporated some comments and justified some arguments. But unfortunately, I am unable to overlook the paper’s methodological shortcomings. For any review paper (systematic or scoping), it is the inclusion-exclusion criteria and, consequently, the selection of studies that provides a result and a scientific conclusion to the research. In the current paper, I could see nearly 7 studies out of 22 (over 30% of studies) do not fall under inclusion criteria. Even if authors are compelled to accept those papers out of their experimental protocol, they should have mentioned it in a separate table with a justification for why they chose it. Besides, the discussion will include only part of those studies, not entirely. For instance, Wolfling et al. 2019 study (mentioned in the paper) has an age range of 17-55 years. It deviates from the inclusion criteria of 0 -18 years. If the authors strongly feel and plan to keep this study (which they did), they have to extract results from only 17- and 18-years old participants from this study. They cannot include this study's complete result (including middle and old age) in this scoping review. This biases the whole review procedure. Similarly, this is the case for all the other studies. Additionally, there are no ways suggested by authors to control these biases.

On the basis of aforesaid arguments, I am afraid that I have to reject the paper and cannot recommend this paper for publication in its current form.